# Fast and Sensitive Quantification of AccQ-Tag Derivatized Amino Acids and Biogenic Amines by UHPLC-UV Analysis from Complex Biological Samples

**DOI:** 10.3390/metabo12030272

**Published:** 2022-03-21

**Authors:** Andrea Guba, Orsolya Bába, József Tőzsér, Éva Csősz, Gergő Kalló

**Affiliations:** 1Proteomics Core Facility, Department of Biochemistry and Molecular Biology, Faculty of Medicine, University of Debrecen, Egyetem tér 1, 4032 Debrecen, Hungary; guba.andrea@med.unideb.hu (A.G.); babaorsi@gmail.com (O.B.); tozser@med.unideb.hu (J.T.); cseva@med.unideb.hu (É.C.); 2Biomarker Research Group, Department of Biochemistry and Molecular Biology, Faculty of Medicine, University of Debrecen, Egyetem tér 1, 4032 Debrecen, Hungary; 3Doctoral School of Molecular Cell and Immune Biology, University of Debrecen, Egyetem tér 1, 4032 Debrecen, Hungary; 4Laboratory of Retroviral Biochemistry, Department of Biochemistry and Molecular Biology, Faculty of Medicine, University of Debrecen, Egyetem tér 1, 4032 Debrecen, Hungary

**Keywords:** derivatization, LC, MS, body fluid, serum, tears, method development, validation

## Abstract

Metabolomic analysis of different body fluids bears high importance in medical sciences. Our aim was to develop and validate a fast UHPLC-UV method for the analysis of 33 amino acids and biogenic amines from complex biological samples. AccQ-Tag derivatization was conducted on target molecules and the derivatized targets were analyzed by UHPLC-UV. The detection of the analytes was carried out with UV analysis and by Selected Reaction Monitoring (SRM)-based targeted mass spectrometry. The method was validated according to the FDA guidelines. Serum and non-stimulated tear samples were collected from five healthy individuals and the samples were analyzed by the method. The method was successfully validated with appropriate accuracy and precision for all 33 biomolecules. A total of 29 analytes were detected in serum samples and 26 of them were quantified. In the tears, 30 amino acids and biogenic amines were identified and 20 of them were quantified. The developed and validated UHPLC-UV method enables the fast and precise analysis of amino acids and biogenic amines from complex biological samples.

## 1. Introduction

In recent years, emerging omics techniques have been widely used for the analysis of complex biological samples as part of system biology studies. In addition to transcriptomics providing information regarding transcription machinery and proteomics concerning protein content, metabolomic analyses involve state of the art techniques for the qualitative and quantitative analysis of the metabolome of the system of interest [1]. Shotgun metabolomic analyses are capable of metabolite profile screening, while with targeted metabolomics, the analysis of several preselected targets is also possible [2]. Metabolomics often relies on analytical techniques such as gas chromatography (GC), liquid chromatography (LC), nuclear magnetic resonance (NMR) and mass spectrometry (MS) [3]. With the help of highly sensitive LC–MS systems, molecules of relatively low concentrations can be analyzed from complex biological samples.

The identification of biomarkers specific for different diseases is an important field in biological and medical sciences. In some conditions, the identified biomarkers have a central role in the homeostatic or pathological processes and their presence or absence plays a key role in the manifestation of disease [4]. In other cases, the presence, absence or altered level of biomarkers is the consequence of the disease with no connection to the cause of the disease [5]. One major problem of biomarker studies is the availability of samples, since highly invasive sample collection methods should be applied in order to obtain biopsy samples from the affected tissues. With the help of omics techniques, the analysis of body fluids which can be collected through non-invasive means, such as tears, sweat, saliva, urine and serum is possible providing an opportunity for the detection of potential new biomarkers for different pathological conditions [6].

Amino acids are the building blocks of proteins and in addition to their role in protein synthesis, they can regulate metabolic pathways as well. Amino acids were found to be potential biomarkers for different pathological conditions: reduced levels of serum tryptophan were identified in patients with depression and other mental disorders [7] and changes in the level of several amino acids have been found in lung, gastric, colorectal, breast and prostate cancers [8]. Moreover, glutamine has been described as bearing an essential role in cancer cell development [9]. In addition to mental disorders and tumors, amino acids also play a role in the pathomechanism of diabetes mellitus, considering that insulin inhibits the degradation of proteins while enhancing amino acid uptake into the cells [8]. Moreover, lysine, aspartate, threonine, methionine and alanine are found to be potential biomarkers for type 2 diabetes mellitus [10].

Biogenic amines are a group of nitrogen-containing small molecules mainly produced via enzymatic decarboxylation of amino acids [11]. Biogenic amines, such as tryptamine, 2-phenethyl amine, putrescine, cadaverine, histamine and tyramine have different physiological functions. Histamine has a role in cell proliferation and differentiation, regeneration and wound healing, vascular permeabilization, neurotransmission; and also an important function in inflammatory reactions [12]. Tryptamine, tyramine and 2-phenethyl amine are called trace amines and have roles in neural and vascular functions [13], while cadaverine and putrescine are important molecules for cell division [14]. In addition to their physiological roles, biogenic amines also play a role in the pathomechanism of several diseases such as hypertension [15], schizophrenia [16] and cancer development [17]. Ethylamine represents an organic compound that was found to be negatively associated with the risk of the development of type 2 diabetes mellitus in a Japanese population [18]. Methylamine is one of the simplest aliphatic amines in the human body and has been identified in many tissues and body fluids [19]. It has been suggested that methylamine plays a role in central nervous system disturbances observed in several pathological conditions [19], and it has been observed that in pregnancy toxemia, the blood level of methylamine remains higher for a longer time after delivery compared to normal pregnancies [20]. Ethanolamine can be found in every cell in the human body as part of phospholipids, and in a free form in body fluids. As the main component of phosphatidylethanolamine, ethanolamine plays a role in neurodegenerative disorders, cancer and ferroptosis [21]. Serotonin as an important neurotransmitter of the central nervous system has a crucial role in the development of depression [22] and has also been found to be associated with obesity and diabetes mellitus [23].

The analysis of amino acids and biogenic amines relies on different biochemical and analytical methods and often requires derivatization. The AccQ-Tag derivatization technique marketed by Waters utilizes 6-aminoquinolyl-N-hydroxysuccinimidyl carbamate to transform primary and secondary amines into highly stable fluorescent derivatives [24]. In the first step of the reaction, the primary and secondary amino groups undergo a reaction with the 6-aminoquinolyl-N-hydroxsuccinimidyl carbamate reagent. The second step is a slower reaction where the excess reagent reacts with water and forms the byproducts 6-aminoquinoline (AMQ), N-hydroxsuccinimide and CO_2_. The last step of the derivatization is a reaction between the major byproduct AMQ and the excess of the reagent forming highly stable bis-aminoquinoline urea. These byproducts do not affect the identification or the quantification of the amino acids [25]. The derivatized amino acids can be separated by liquid chromatography and the fluorescent derivates can be detected photometrically at 260 nm.

Although the analysis of amino acids and biogenic amines bears relevance in the medical sciences, to the best of our knowledge, there is no analytical method for the simultaneous analysis of these molecules. In this study, we have developed and validated a UHPLC-UV technique coupled with mass spectrometry analysis for the rapid and simultaneous analysis of 33 AccQ-Tag derivatized amino acids and biogenic amines from complex biological samples. The selection of the biogenic amines for this study was based on their roles in physiological processes and in pathological conditions. The validated method was tested on serum and tear samples proving the reliability of the method in medical sciences.

## 2. Results and Discussion

### 2.1. UHPLC–MS Method Development

Considering that the level of amino acids and biogenic amines can indicate pathological changes, we have developed a rapid UHPLC-UV method coupled with mass spectrometry analysis for the simultaneous analysis of AccQ-Tag derivatized alanine, arginine, asparagine, aspartate, cadaverine, citrulline, cysteine, ethanolamine, ethylamine, glutamate, glutamine, glycine, histamine, histidine, isoleucine, leucine, lysine, methionine, methylamine, ornithine, phenylalanine, proline, putrescine, serine, serotonin, taurine, threonine, tryptamine, tryptophan, tyramine, tyrosine, valine and 2-phenethyl-amine.

The method was developed on an Acquity H-class UHPLC system coupled with a 5500QTRAP tandem mass spectrometer. The elution profile was optimized and a rapid 11 min gradient capable for the separation of the 33 molecules of interest was developed. The derivatized molecules were detected with a photodiode array (PDA) detector at 260 nm wavelength and with the 5500QTRAP mass spectrometer in positive Selected Reaction Monitoring (SRM) mode. Figure 1 shows the representative chromatogram of the derivatized molecules registered by the UHPLC-UV analysis.

Using the applied settings, the separation of the derivatized molecules was successful except for tryptamine and 2-phenethyl-amine. Therefore, the individual analysis of these two molecules was conducted through SRM analyses (Figure 2).

Histidine, citrulline, ornithine, serotonin, cadaverine and putrescine have two available amine groups for derivatization, thus, during the method development the SRM parameters were selected for the detection of both the single and double derivatized forms. According to our results, citrulline contained one derivatized amine group while in the case of ornithine, we observed double derivatized end products. In the case of serotonin, both single and double derivatized peaks were detected by mass spectrometry while only the single-derivatized form was detected by UHPLC-UV. In the case of lysine, cadaverine and putrescine both single and double derivatized peaks were detected, but since the intensity of the single derivatized peaks was low and the ratio between the single and double derivatized forms was constant, we used the double derivatized peaks for quantification. The appropriate SRM transition was collected into a method file and by using the retention time of each analyte, a scheduled SRM method was created for the analysis of the selected molecules.

### 2.2. UHPLC–MS Method Validation

The developed method was further validated based on the work of Galba et al. and Grey et al. [26,27] in accordance with the FDA guidelines. Calibration curves of 0.25–30 µmol/L range were registered in MilliQ water, serum and tear matrices and the linear range, limit of detection (LOD) and limit of quantification (LOQ) were calculated based on the results obtained by LC (Table 1 and Appendix A) and by SRM analysis (Appendix A).

Based on the results obtained by UHPLC-UV analysis, the method exhibited good sensitivity and a wide dynamic range; features, which can be important for the analysis of the 33 target molecules in biological samples. The comparison of the results registered by UHPLC-UV and by SRM analyses revealed that while the individual analysis of tryptamine and 2-phenethyl-amine can be performed by SRM analysis, the calculated linear dynamic ranges were wider in the case of the UHPLC-UV analysis compared to SRM. Considering the obtained data, we decided to use the UHPLC-UV analysis for quantification of the derivatized amino acids and biogenic amines and for further validation of the developed method. SRM analysis was employed for the confirmation of the chromatographic peaks and for the qualitative and quantitative analysis of co-eluting tryptamine and 2-phenethyl-amine.

For validation of the developed method, quality control (QC) samples were generated in MilliQ water, serum and tear matrices, respectively, by spiking them with 2.5 µmol/L, 7.5 µmol/L and 15 µmol/L of each analyte. The intra- and inter-day accuracy and precision of the method were investigated by the analysis of QC samples and the obtained results are summarized in Appendix A. The accuracy of the 33 derivatized molecules ranged from 85.66% to 114.41% in MilliQ water, from 90.92% to 114.12% in serum matrix and from 91.39% to 114.80% in tear matrix. The intra-day and inter-day precisions were below 15% in MilliQ water and tear matrix, and below 9% in serum matrix. The registered results meet the FDA criteria for acceptable accuracy (±15% of nominal concentration) and precision (±15% RSD) indicating the usefulness of the method in the analysis of the selected 33 biomolecules in complex biological samples. The recovery calculated from the data obtained by the analysis of QC samples was higher than 85% (Appendix A). The matrix effect was tested (Appendix A) revealing a considerable matrix effect. For the appropriate analysis of these molecules in serum and tears, it is recommended that the calibration curve be prepared in the appropriate matrix. The stability of the molecules under different conditions, such as in the autosampler and during freeze–thaw cycles before and after derivatization was investigated as well (Appendix A). Considering our data, the samples should not be stored in the autosampler over 12 h and freeze–thaw cycles should be avoided in order to obtain precise quantification data.

The data altogether indicated appropriate linear range and sensitivity for the analysis of the selected 33 biomolecules which can be useful for the examination of clinical samples.

### 2.3. Amino Acid and Biogenic Amine Content of Serum and Tears

With the help of the validated method, we determined the concentration of the selected amino acids and biogenic amines in serum and tear samples of healthy volunteers. After sample collection, serum and tears were purified on 3 kDa cut-off spin columns and were subjected to AccQ-tag derivatization. Sample loss was examined by the analysis of amino acid mixtures before and after purification, and a significant difference was not identified between the peak areas of purified and non-purified samples (Appendix A). The derivatized samples were analyzed and the concentration of the molecules of interest was calculated. Regarding the fluids of the human body, serum represents the most commonly used biological material in the medical sciences for the diagnosis of different pathological changes and for the follow-up of applied therapies. Serum is a protein and metabolite rich body fluid that can be obtained in a minimally-invasive manner and is an important source for biomarker analyses [28,29].

In the serum samples obtained from 5 healthy volunteers we could identify all the derivatized amino acids and biogenic amines except histamine, tyramine, tryptamine and 2-phenethyl amine (Table 2). Ethylamine was detected with UHPLC-UV analysis in the serum samples, but the concentration was below the limit of quantification, while methylamine was only detected by SRM analysis. Histamine, tyramine, tryptamine and 2-phenethyl amine were not detected in the samples.

The mean concentration of the quantified biomolecules in serum is shown in Figure 3.

Based on the results, glutamine and alanine have the highest concentration in serum in agreement with the scientific literature [30,31,32]. In addition to the major nitrogen transporters, the concentrations of serine, glycine, threonine, proline, lysine, valine and leucine were found to be greater than 100 µmol/L. The level of the studied biogenic amines was considerably lower in the serum compared to the level of proteinogenic and non-proteinogenic amino acids.

Monitoring the changes in the amino acid profile of the serum has relevance in biological and medical sciences due to the fact, that amino acids possess significant roles in the homeostasis and pathogenesis of several diseases. The serum level of glycine was found to be associated with insulin resistance and metabolic syndrome [33,34]. The level of branched chain amino acids such as valine, leucine and isoleucine is elevated in cases of Maple Syrup Urine disease [35], and significantly decreased in autism caused by branched chain keto acid dehydrogenase kinase deficiency [36]. Ethanolamine as a part of plasmalogens can bear influence on the development of neurological problems, such as Alzheimer’s disease [37,38]. Polyamines, such as putrescine and cadaverine have various functions in the human body and have relevance in pathological conditions, such as chronic renal failure or autoimmune thyroid diseases [39,40]. Moreover, cadaverine was identified as a regulator of breast cancer development [41].

While serum is the most commonly used body fluid for the diagnosis of various disorders, tears also have a high impact in medical sciences [42,43,44,45,46]. Moreover, tears are metabolite and protein rich body fluids and although it is not a common biomaterial for diagnostic purposes, it is an invaluable source for biomarker studies [6,47,48,49]. Another benefit of tear analysis is the non-invasive and relatively simple sample collection possibility.

Tear samples were collected from 8 eyes of the same 5 healthy volunteers and after the derivatization and analysis, the detection of the molecules of interest was possible except serotonin, tryptamine and 2-phenetyl amine (Table 3). We attempted to collect tear samples from both eyes and if it was possible, we considered them as individual samples. Ethanolamine, ethylamine, methionine, cadaverine, tyramine, leucine and phenylalanine were detected in the tear samples with UHPLC-UV analysis, but the concentration was below the limit of quantification. Methylamine was only detected by SRM analysis and putrescine was not detected in the analyzed tear samples.

The mean concentration of the quantified biomolecules in tears is shown in Figure 4.

The quantification of 17 proteinogenic and 3 non-proteinogenic amino acids was possible with the developed UHPLC-UV method coupled with SRM analysis. Similar to the results obtained by the analysis of serum samples, in tears the concentration of biogenic amines was lower than that of the amino acids. Our results show that serine exhibits the highest concentration in tears among the analyzed biomolecules. We also demonstrated that the concentration of the analyzed amino acids and biogenic amines are different in tears compared to serum. The changes in the metabolite profile of tears are not as well studied as that of serum. Studies have shown that changes in the metabolomic profile of tears were observed in keratoconus, keratitis and dry eye disease [50,51,52]. However, limited information is available regarding the importance of amino acids and biogenic amines in tears. ChenZhuo et al. demonstrated that the level of alanine, aspartate and taurine was significantly higher in the tears of patients with dry eye disease compared to healthy controls [53] and Nakatsukasa et al. found reduced arginine, methionine and taurine, and increased ornithine, lysine and threonine levels in the tears of patients with severe ocular surface diseases [54]. While there is limited information in the scientific literature, the few studies available revealed that the analysis of the amino acid and biogenic amine profile of tears can provide valuable information regarding the molecular changes characteristic of different disorders.

In this article, we have presented a fast and sensitive validated UHPLC-UV method coupled with SRM analysis for the simultaneous analysis of 33 AccQ-Tag derivatized amino acids and biogenic amines. The method was found to be accurate, sensitive and precise and meets the FDA criteria. We also tested the method on serum and tear samples and the results indicated that with the developed assay, the amino acids and biogenic amines can be identified and quantified simultaneously in complex biological samples. The changes in the amino acid and biogenic amine profile of these body fluids can indicate alterations of the homeostatic functions or can reflect different pathological conditions.

While the method was validated and found to be useful for the analysis of body fluids, the sample collection and preparation might be challenging. Our results indicated that the freeze–thaw cycles should be avoided during sample preparation.

A possible limitation can be the variability of the samples and the low volume of basal tears collected. The tear production rate shows high variability between individuals and in many cases, tear collection is not possible. Our method requires 3 µL of tear samples, therefore, subjects with low tear secretion rates may not be involved in studies using the presented method. We have demonstrated high variability of the amino acid content of tears not just between individuals but between the left and right eye as well. During tear collection, the same procedure should be carried out preferably by the same person in order to decrease the variance between the collected samples, but the individual variance between the composition of tears will still present an issue. In addition to the presented body fluids, the method can be further optimized for the analysis of other body fluids with high medical impact, such as saliva and sweat. Another limitation was that the concentration of biogenic amines remained mostly under the limit of quantification. While the concentration of these biogenic amines in healthy conditions is low in body fluids, pathological changes such as diabetes can increase the concentration of several molecules such as putrescine, histamine, etc. [55,56]. The presented method can be useful for the examination of the molecular background of diseases bearing high importance in medical sciences.

## 3. Materials and Methods

### 3.1. Chemicals and Reagents

AccQ-Tag Ultra derivatization kit, a mixture of 17 proteinogenic amino acids (histidine, serine, arginine, glycine, aspartate, glutamate, alanine, threonine, proline, cysteine, lysine, tyrosine, methionine, valine, isoleucine, leucine, and phenylalanine) and AccQ-Tag Ultra eluent A and B were purchased from Waters (Milford, MA, USA). LC–MS grade water, 3kDa Nanosep columns and asparagine, glutamine, taurine, histamine, ethanolamine, methylamine, citrulline, ethylamine, ornithine, putrescine, serotonin, cadaverine, tyramine, tryptamine and 2-phenethyl-amine standards were obtained from Sigma-Aldrich (St. Louis, MI, USA). Stable isotope-labelled (SIL) tryptophan was purchased from Cambridge Isotope Laboratories (Tewksbury, MA, USA).

### 3.2. Preparation of Standard Solutions and QC Samples

The stock solution of the amino acids and biogenic amines was prepared by spiking the Waters amino acid mixture with asparagine, glutamine, SIL tryptophan and the biogenic amines to obtain a 2.5 mmol/L final concentration for each analyte. The stock solution was stored at −20 °C. The calibration standards were prepared from the stock solution through serial dilutions.

The QC samples were prepared in MilliQ water, serum and tear matrices, respectively, by spiking them either with 2.5 µmol/L, 7.5 µmol/L or 15 µmol/L of each analyte. QC samples were used for the analysis of the recovery, matrix effect, intra- and inter-day precision and stability.

### 3.3. UHPLC–MS and Data Analysis

Prior to UHPLC–MS analysis, samples were derivatized using AccQ-Tag Ultra derivatization reagent according to the manufacturer’s instructions. Briefly, 10 µL sample was mixed with 20 µL AccQ-tag Ultra derivatization reagent and with 70 µL borate buffer (pH 8.8) and incubated at 55 °C for 10 min [24].

Liquid chromatographic separation was performed on an Acquity H-class UPLC system (Waters, Milford, MA, USA) controlled by the Empower 3 software (Waters, Milford, MA, USA). The separation of the derivatized amino acids and biogenic amines was carried out on an AccQ-tag Ultra C18 column (1.7 µm; 2.1 × 100 mm, Waters, Milford, MA, USA) guarded by an Acquity in-line filter (0.2 µm; 2.1 mm, Waters, Milford, MA, USA). An 11 min long gradient was established with 0.65 mL/min flow rate and 54 °C column temperature. Solvent A was 100% AccQ-tag Ultra eluent A, solvent B was 10% AccQ-tag Ultra eluent B in LC–MS grade water, solvent C was LC water and solvent D was 100% AccQ-tag Ultra eluent B. Table 4 contains the elution profile of the UPLC separation.

The PDA detector of the instrument was set to 260 nm wavelength with 10 points/s sampling rate.

SRM-based targeted mass spectrometry analyses were carried out on a 5500QTRAP (ABSciex, Framingham, MA, USA) mass spectrometer controlled by the Analyst software (version 1.6.3., ABSciex, Framingham, MA, USA). The eluates from the LC column were ionized using electrospray ionization with 5500 V spray voltage and the positive ion mode SRM spectra were recorded. Other acquisition parameters were as follows: the ion source gas 1 was set to 30 psi; the ion source gas 2 was set to 50 psi; the curtain gas was set to 30 psi, and the source temperature was 500 °C. The detailed parameters of the SRM experiment are presented in Table 5.

The registered chromatograms were analyzed with the Empower 3 software and the SRM spectra were analyzed with the Skyline software [57]. The acquired SRM data were uploaded to the Panorama website (https://panoramaweb.org/University%20of%20Debrecen/AA_BA_Validation/project-begin.view? accessed on 21 January 2022) and are publicly available.

### 3.4. Method Validation

The UHPLC-UV method coupled with SRM analysis was validated for linearity, accuracy, intra- and inter-day variability, recovery, matrix effect, limit of detection (LOD), limit of quantification (LOQ) and stability according to the FDA guidelines [58] as described by Galba et al. and Grey et al. [26,27].

The linear range for all studied analytes was determined by analyzing a concentration range of 0.25–30 µmol/L in MilliQ water, serum and tear matrices, respectively, in triplicates. Weighted (1/x) least-squares linear regression fit was applied for the result obtained by LC analysis and by SRM. Each calibration sample was measured in triplicates. The LOD and LOQ values were calculated from the calibration curves based on the slope (m) and the standard deviation of the slope (SD_m_) as follows:LOD = 3 × SD_m_/m (1)
LOQ = 10 × SD_m_/m (2)

The accuracy, recovery, matrix effect, intra- and inter-day variability and stability were calculated from 5 injections of each QC sample (2.5 µmol/L, 7.5 µmol/L and 15 µmol/L). The inter-day variability and accuracy of the method were determined by the analysis of the QC samples in the three matrices over 3 consecutive days. The recovery was evaluated by the comparison of the determined QC concentration with their nominal concentrations. The matrix effect (ME) was calculated based on the following formula:ME = [(A − B)/A] × 100 (3)
where A is the peak area of the analytes in MilliQ water and B is the peak area of the same concentration of the analytes spiked into serum or tear matrices. The autosampler stability was tested by the analysis of QC samples stored in the autosampler at 4 °C for 12 h. The freeze–thaw stability of the samples was tested by the analysis of the QC samples after three freeze–thaw cycles at −70 °C before and after AccQ-Tag derivatization. The stability was calculated by the comparison of the concentrations after the freeze–thaw cycles with the concentrations of freshly prepared QC samples.

### 3.5. Analysis of Tears and Serum Samples

Serum and tears were collected from 5 healthy volunteers. Sample collection complied with the guidelines of the Helsinki Declaration and ethical approval was obtained from the University of Debrecen Ethics Committee (DEOEC RKEB/IKEB 4701A-2016).

Tear collection was conducted using sterile glass capillary tubes (VWR Ltd., Radnor, PA, USA) without local anesthesia or stimulation [59]. Non-stimulated tear samples were collected from both eyes of each subject as described before [60]. A 3 µL individual tear sample was supplemented with 47 µL MilliQ water and transferred to Nanosep3K (Sigma-Aldrich, St. Louis, MI, USA) size-exclusion spin columns in order to separate the amino acids from the proteins and molecules with higher molecular weight. Samples were centrifuged with 16.000× *g* at 4 °C for 2 × 10 min and the flowthroughs were dried in speed-vac. The dried samples were dissolved in borate buffer and subjected to AccQ-Tag Ultra derivatization according to the manufacturer’s recommendation.

A 100 µL serum sample was transferred to the Nanosep3K spin columns and then centrifuged with 16.000× *g* at 4 °C for 2 × 10 min, and then 10 µL flow through was subjected to AccQ-Tag Ultra derivatization.

Serum samples were analyzed with 10-fold dilution and without dilution, while tear samples were injected into the LC–MS system without dilution. In all cases, 1 µL sample was injected into the LC column in duplicates. Amino acid concentration of the samples was calculated for the original sample volume.

## 4. Conclusions

In this paper, we present a rapid and sensitive UHPLC-UV method coupled with mass spectrometry for the simultaneous analysis of AccQ-Tag derivatized alanine, arginine, asparagine, aspartate, cadaverine, citrulline, cysteine, ethanolamine, ethylamine, glutamate, glutamine, glycine, histamine, histidine, isoleucine, leucine, lysine, methionine, methylamine, ornithine, phenylalanine, proline, putrescine, serine, serotonin, taurine, threonine, tryptamine, tryptophan, tyramine, tyrosine, valine and 2-phenethyl-amine. The accuracy and precision of the developed method is good and meets the FDA criteria. Moreover, we demonstrated the utility of the developed method for the analysis of body fluids such as serum and tears. By applying this method, changes in the amino acid and biogenic amine profiles of serum and tears can be analyzed rendering this method a useful tool for the analysis of body fluids.

## Figures and Tables

**Figure 1 metabolites-12-00272-f001:**
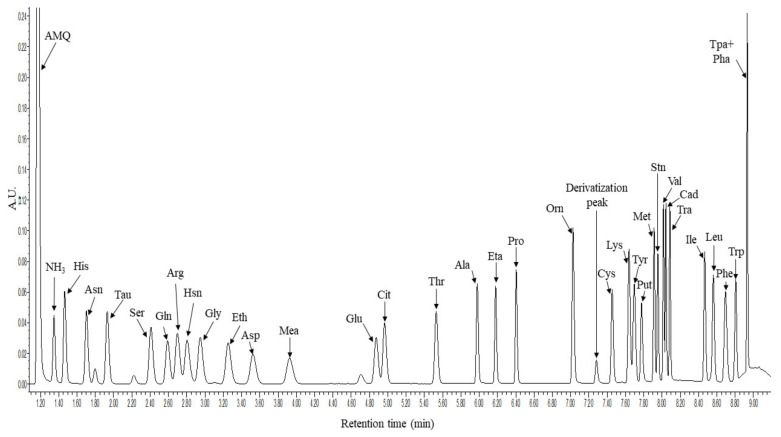
Representative chromatogram of the 33 derivatized amino acids and biogenic amines. The *x* axis shows the retention time in minutes while the *y* axis shows the intensity in form of arbitrary units (AU) registered in the PDA detector at 260 nm. For amino acids the three-letter code was used, AMQ: 6-aminoquinoline, Tau: taurine, Hsn: histamine, Eth: ethanolamine, Mea: methylamine, Eta: ethylamine, Derivatization peak: bis-aminoquinoline urea, Put: putrescine, Stn: serotonin, Cad: cadaverine, Tra: tyramine, Tpa + Pha: tryptamine and 2-phenethly-amine.

**Figure 2 metabolites-12-00272-f002:**
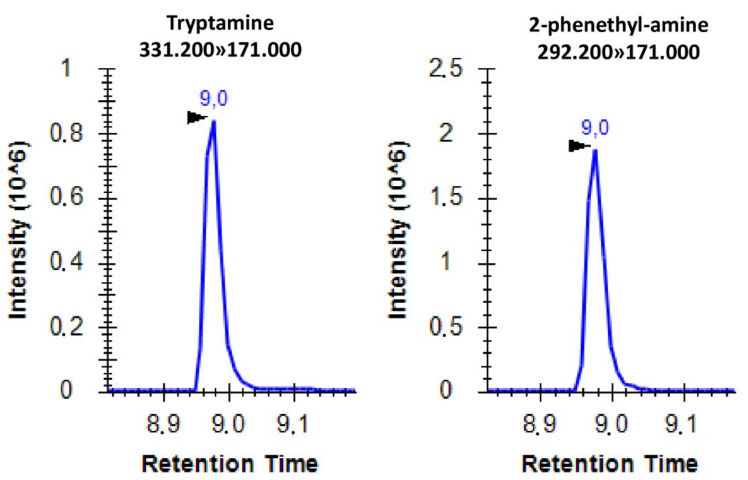
Representative SRM spectrum of AccQ-Tag derivatized tryptamine and 2-phenethyl-amine with the corresponding SRM transitions. The *x* axis shows the retention time in minutes while the *y* axis shows the peak intensity.

**Figure 3 metabolites-12-00272-f003:**
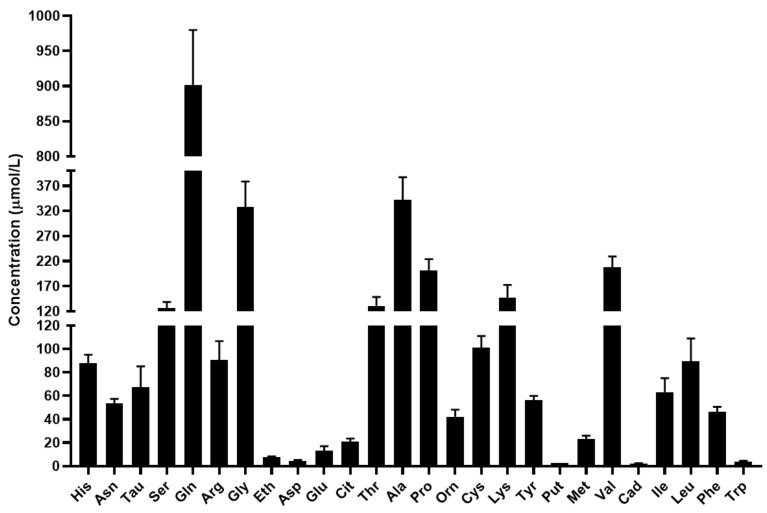
Concentration of the quantified amino acids and biogenic amines in serum in µmol/L. For amino acids the three-letter code was used, Tau: taurine, Eth: ethanolamine, Put: putrescine, Cad: cadaverine, Tra: tyramine.

**Figure 4 metabolites-12-00272-f004:**
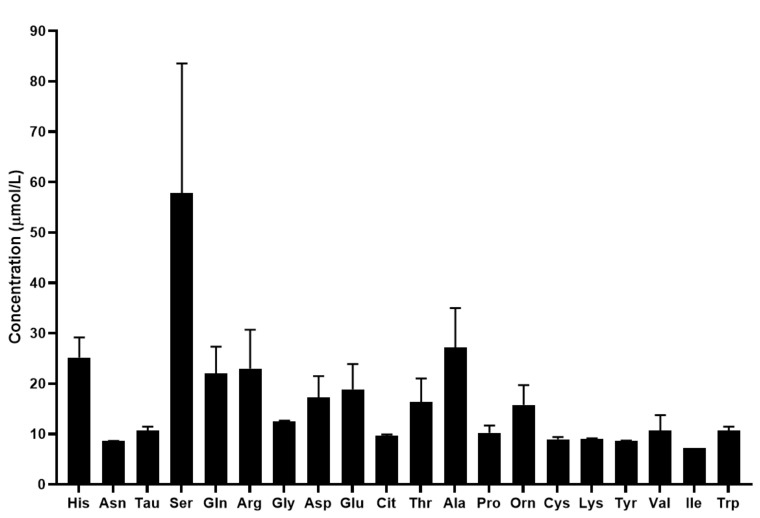
Concentration of the quantified amino acids and biogenic amines in tears in µmol/L. For amino acids the three-letter code was used, Tau: taurine.

**Table 1 metabolites-12-00272-t001:** Calibration parameters of the analyzed molecules in different matrices obtained with UHPLC-UV analysis. t_R_: retention time, MQ: MilliQ water, LOD: limit of detection, LOQ: limit of quantification. The data represent the mean value of three technical replicates.

Compound	Abbreviation	t_R_ (min)			LOD (µmol/L)			LOQ (µmol/L)			Linear Range (µmol/L)		
		MQ	Serum	Tears	MQ	Serum	Tears	MQ	Serum	Tears	MQ	Serum	Tears
Histidine	His	1.445	1.445	1.447	1.81	2.30	0.73	6.04	7.68	2.42	1.00–25.00	2.50–30.00	1.00–25.00
Asparagine	Asn	1.678	1.677	1.679	0.38	0.34	0.59	1.27	1.12	1.98	0.50–25.00	0.50–30.00	0.50–20.00
Taurine	Tau	1.916	1.917	1.915	0.20	0.20	0.22	0.68	0.66	0.73	0.25–25.00	2.50–30.00	0.50–25.00
Serine	Ser	2.370	2.367	2.372	0.21	0.22	0.18	0.69	0.74	0.59	0.25–25.00	2.50–25.00	0.50–25.00
Glutamine	Gln	2.547	2.546	2.550	2.02	1.91	1.67	6.74	6.37	5.58	0.50–25.00	0.50–30.00	1.00–25.00
Arginine	Arg	2.643	2.643	2.650	1.37	0.10	0.78	4.57	0.32	2.60	0.50–20.00	0.50–25.00	1.00–25.00
Histamine	Hsn	2.748	2.744	2.754	0.72	0.75	0.52	2.39	2.51	1.73	0.50–25.00	2.50–25.00	0.50–25.00
Glycine	Gly	2.911	2.906	2.913	0.41	0.38	0.74	1.36	1.26	2.45	0.50–25.00	2.50–30.00	1.00–25.00
Ethanolamine	Eth	3.208	3.205	3.212	0.25	0.22	0.28	0.84	0.75	0.93	0.50–25.00	2.50–30.00	0.25–15.00
Aspartate	Asp	3.482	3.475	3.481	0.25	0.22	0.20	0.84	0.74	0.68	0.50–25.00	0.50–30.00	0.50–20.00
Methylamine	Mea	3.876	3.870	3.879	0.26	0.31	0.20	0.85	1.02	0.68	0.25–25.00	0.50–30.00	0.25–15.00
Glutamate	Glu	4.845	4.840	4.841	0.44	0.49	0.36	1.48	1.64	1.20	0.25–25.00	0.50–30.00	0.25–15.00
Citrulline	Cit	4.940	4.938	4.938	0.37	0.44	0.28	1.24	1.46	0.92	0.25–25.00	2.50–30.00	0.25–15.00
Threonine	Thr	5.493	5.500	5.514	0.18	0.17	0.20	0.61	0.56	0.66	0.25–25.00	2.50–30.00	1.00–25.00
Alanine	Ala	5.946	5.956	5.977	0.22	0.24	0.12	0.74	0.78	0.41	0.25–25.00	1.00–30.00	0.25–15.00
Ethylamine	Eta	6.146	6.157	6.181	0.18	0.13	0.15	0.61	0.44	0.50	0.50–25.00	0.25–30.00	0.25–15.00
Proline	Pro	6.356	6.378	6.417	0.13	0.16	0.26	0.44	0.53	0.88	0.25–25.00	2.50–30.00	1.00–25.00
Ornithine	Orn	6.914	6.970	7.061	0.15	0.17	0.16	0.50	0.56	0.55	0.25–25.00	2.50–30.00	0.25–15.00
Cysteine	Cys	7.336	7.397	7.492	0.17	0.14	0.20	0.56	0.47	0.67	0.50–25.00	2.50–30.00	1.00–25.00
Lysine	Lys	7.521	7.578	7.668	0.23	0.26	0.52	0.75	0.85	1.74	0.25–25.00	2.50–30.00	0.25–15.00
Tyrosine	Tyr	7.596	7.643	7.724	0.17	0.24	0.78	0.56	0.81	2.60	0.50–25.00	2.50–30.00	0.50–25.00
Putrescine	Put	7.665	7.719	7.796	0.14	0.26	0.13	0.47	0.87	0.44	0.25–25.00	2.50–30.00	0.25–15.00
Methionine	Met	7.865	7.887	7.922	0.13	0.13	0.38	0.44	0.43	1.28	0.50–25.00	2.50–30.00	1.00–25.00
Serotonin	Stn	7.920	7.935	7.962	0.81	0.12	0.38	2.71	0.41	1.28	0.50–30.00	2.50–25.00	0.25–30.00
Valine	Val	7.991	8.001	8.019	0.11	0.39	0.21	0.38	1.30	0.69	0.25–25.00	2.50–30.00	0.25–20.00
Cadaverine	Cad	8.029	8.033	8.044	0.201	0.14	0.33	0.69	0.47	1.10	0.25–25.00	2.50–30.00	0.25–15.00
Tyramine	Tra	8.069	8.075	8.090	0.30	0.30	0.19	0.99	0.99	0.65	0.25–25.00	2.50–30.00	0.25–20.00
Isoleucine	Ile	8.446	8.453	8.468	0.13	0.17	0.18	0.43	0.55	0.61	0.25–25.00	2.50–30.00	0.50–20.00
Leucine	Leu	8.535	8.544	8.562	0.11	0.11	0.24	0.36	0.38	0.82	0.25–20.00	0.50–25.00	0.25–20.00
Phenylalanine	Phe	8.662	8.673	8.696	0.17	0.16	0.21	0.55	0.53	0.70	0.25–25.00	2.50–30.00	0.50–25.00
Tryptophan	Trp	8.785	8.794	8.807	0.61	0.22	0.20	2.02	0.72	0.66	1.00–25.00	2.50–30.00	0.25–25.00
Tryptamine + 2-phenethyl-amine	Tpa + Pha	8.932	8.931	8.933	0.14	0.16	0.27	0.47	0.58	0.89	0.25–25.00	2.50–30.00	0.25–25.00

**Table 2 metabolites-12-00272-t002:** Concentration of amino acids and biogenic amines in serum. <LOQ: the molecule was identified with UHPLC-UV analysis, but the concentration was below the limit of quantification, SRM: the molecule was detected by SRM, but the concentration was below the limit of quantification, N.D.: not detected. The data represent the mean value of two technical replicates.

Compound	Abbreviation	Concentration (µmol/L)				
		Subject 1	Subject 2	Subject 3	Subject 4	Subject 5
Histidine	His	70.96	102.66	103.64	77.36	83.75
Asparagine	Asn	42.47	64.76	50.61	55.29	53.83
Taurine	Tau	52.09	129.92	41.72	53.22	60.30
Serine	Ser	92.28	142.17	104.86	145.38	144.03
Glutamine	Gln	790.05	702.40	1025.00	919.45	1073.15
Arginine	Arg	46.95	115.61	72.63	126.12	92.15
Glycine	Gly	389.10	161.67	377.60	298.28	410.85
Ethanolamine	Eth	7.97	6.96	7.81	8.19	8.75
Aspartate	Asp	2.67	7.13	<LOQ	3.14	4.07
Methylamine	Mea	SRM	SRM	SRM	SRM	SRM
Glutamate	Glu	7.93	24.02	5.02	14.91	14.53
Citrulline	Cit	16.03	23.33	19.19	28.84	17.85
Threonine	Thr	80.53	140.02	144.87	114.91	174.50
Alanine	Ala	324.90	346.20	204.81	376.20	455.50
Ethylamine	Eta	<LOQ	<LOQ	<LOQ	<LOQ	<LOQ
Proline	Pro	277.47	173.13	167.17	195.59	194.99
Ornithine	Orn	28.09	55.65	39.04	52.83	35.86
Cysteine	Cys	103.82	82.85	92.39	92.79	133.90
Lysine	Lys	103.09	223.05	129.16	111.65	171.02
Tyrosine	Tyr	48.08	62.98	56.47	63.87	50.67
Putrescine	Put	<LOQ	<LOQ	<LOQ	<LOQ	2.63
Methionine	Met	17.89	26.42	31.44	18.52	21.25
Serotonin	Stn	SRM	SRM	SRM	SRM	SRM
Valine	Val	204.36	282.25	188.30	190.14	174.23
Cadaverine	Cad	<LOQ	2.41	<LOQ	N.D.	N.D.
Isoleucine	Ile	53.77	104.82	56.57	56.86	43.73
Leucine	Leu	75.31	158.39	76.30	73.11	65.44
Phenylalanine	Phe	46.62	59.21	36.84	48.95	39.50
Tryptophan	Trp	<LOQ	6.33	2.60	2.915	2.27

**Table 3 metabolites-12-00272-t003:** Concentration of amino acids and biogenic amines in tears. OS: Oculus sinister, OD: oculus dexter, <LOQ: the molecule was identified with UHPLC-UV analysis, but the concentration was lower than the limit of quantification, SRM: the molecule was detected by SRM, but the concentration was below the limit of quantification, N.D.: not detected. The data represent the mean value of two technical replicates.

Compound	Abbreviation	Concentration (µmol/L)							
		Subject 1		Subject 2		Subject 3		Subject 4	Subject 5
		OS	OD	OS	OD	OS	OD	OD	OD
Histidine	His	19.55	30.85	<LOQ	<LOQ	N.D.	N.D.	N.D.	N.D.
Asparagine	Asn	<LOQ	8.65	<LOQ	<LOQ	SRM	SRM	SRM	SRM
Taurine	Tau	7.77	12.47	10.38	11.68	9.67	11.48	11.38	<LOQ
Serine	Ser	103.98	149.07	64.47	83.27	26.37	16.60	10.25	9.03
Glutamine	Gln	20.97	38.72	9.70	17.77	<LOQ	23.15	<LOQ	<LOQ
Arginine	Arg	36.98	46.60	18.80	17.52	10.25	<LOQ	<LOQ	7.70
Histamine	Hsn	N.D.	N.D.	N.D.	N.D.	SRM	N.D.	N.D.	N.D.
Glycine	Gly	<LOQ	12.58	<LOQ	<LOQ	SRM	SRM	SRM	SRM
Ethanolamine	Eth	<LOQ	<LOQ	<LOQ	<LOQ	<LOQ	<LOQ	<LOQ	<LOQ
Aspartate	Asp	13.35	26.98	<LOQ	11.53	<LOQ	<LOQ	<LOQ	<LOQ
Methylamine	Mea	SRM	SRM	SRM	SRM	SRM	SRM	SRM	SRM
Glutamate	Glu	15.35	30.22	<LOQ	11.02	<LOQ	<LOQ	<LOQ	<LOQ
Citrulline	Cit	9.33	10.03	<LOQ	<LOQ	<LOQ	<LOQ	<LOQ	<LOQ
Threonine	Thr	21.53	31.72	13.40	16.712	7.68	7.167	<LOQ	<LOQ
Alanine	Ala	40.02	48.17	25.53	30.00	10.45	9.05	<LOQ	<LOQ
Ethylamine	Eta	<LOQ	<LOQ	<LOQ	<LOQ	<LOQ	<LOQ	<LOQ	<LOQ
Proline	Pro	7.43	13.22	<LOQ	<LOQ	<LOQ	10.07	<LOQ	<LOQ
Ornithine	Orn	11.12	24.93	11.23	<LOQ	<LOQ	<LOQ	<LOQ	<LOQ
Cysteine	Cys	9.32	9.33	8.75	8.57	10.57	7.93	<LOQ	8.33
Lysine	Lys	9.10	<LOQ	<LOQ	<LOQ	<LOQ	<LOQ	<LOQ	<LOQ
Tyrosine	Tyr	8.62	<LOQ	<LOQ	<LOQ	<LOQ	<LOQ	<LOQ	<LOQ
Methionine	Met	<LOQ	<LOQ	<LOQ	<LOQ	<LOQ	<LOQ	N.D.	<LOQ
Serotonin	Stn	N.D.	N.D.	N.D.	N.D.	SRM	N.D.	N.D.	N.D.
Valine	Val	9.88	14.67	<LOQ	7.48	<LOQ	<LOQ	<LOQ	<LOQ
Cadaverine	Cad	N.D.	<LOQ	<LOQ	<LOQ	<LOQ	<LOQ	N.D.	N.D.
Tyramine	Tra	<LOQ	<LOQ	N.D.	<LOQ	<LOQ	<LOQ	<LOQ	<LOQ
Isoleucine	Ile	7.30	<LOQ	<LOQ	<LOQ	<LOQ	<LOQ	<LOQ	<LOQ
Leucine	Leu	<LOQ	<LOQ	<LOQ	<LOQ	<LOQ	<LOQ	<LOQ	<LOQ
Phenylalanine	Phe	<LOQ	<LOQ	<LOQ	<LOQ	<LOQ	<LOQ	<LOQ	<LOQ
Tryptophan	Trp	8.88	<LOQ	<LOQ	<LOQ	<LOQ	<LOQ	<LOQ	<LOQ
Tryptamine + 2-phenethyl-amine	Tpa + Pha	N.D.	N.D.	N.D.	N.D.	SRM	N.D.	N.D.	N.D.

**Table 4 metabolites-12-00272-t004:** Elution profile parameters. Solvent A: 100% AccQ-tag Ultra eluent A, solvent B: 10% AccQ-tag Ultra eluent B in LC–MS grade water, solvent C: LC–MS grade water, solvent D: 100% AccQ-tag Ultra eluent B.

Time (min)	Flow Rate (mL/min)	Solvent A (%)	Solvent B (%)	Solvent C (%)	Solvent D (%)	Curve
0.00	0.65	10.00	0.00	90.00	0.00	initial
0.29	0.65	9.90	0.00	90.10	0.00	6
3.50	0.65	9.90	0.00	90.10	0.00	6
4.60	0.65	9.90	25.00	65.10	0.00	7
5.49	0.65	9.00	80.00	11.00	0.00	6
7.10	0.65	8.00	25.00	57.90	9.10	6
7.30	0.65	8.00	15.60	57.90	18.50	6
7.50	0.65	8.00	12.00	57.90	22.10	6
8.20	0.65	7.80	0.00	77.20	15.00	6
8.30	0.65	4.00	0.00	36.30	59.70	6
8.55	0.65	4.00	0.00	36.30	59.70	6
8.60	0.65	4.00	65.00	26.00	5.00	6
9.20	0.65	4.00	60.00	36.00	0.00	6
9.70	0.65	10.00	0.00	90.00	0.00	6
10.90	0.65	10.00	0.00	90.00	0.00	6

**Table 5 metabolites-12-00272-t005:** Parameters of the Selected Reaction Monitoring (SRM) analysis. Q1 *m*/*z*: Parent ion, Q3 *m*/*z*: Fragment ion, t_R_ window: Retention time window in minutes, DP: Declustering potential in eV, CE: Collision energy in eV. SIL tryptophan: Stable isotope-labelled tryptophan.

Compound	Q1 (*m*/*z*)	Q3 (*m*/*z*)	t_R_ Window (min)	DP (eV)	CE (eV)
Histidine	326.00	171.00	1.06–1.90	230	18
Asparagine	303.00	171.00	1.30–2.14	150	15
Taurine	296.15	171.00	1.51–2.35	120	25
Serine	276.00	171.00	2.00–2.84	230	18
Glutamine	317.00	171.00	2.16–3.00	210	16
Arginine	345.00	171.00	2.28–3.12	110	21
Histamine	282.15	171.00	2.39–3.23	120	25
Glycine	246.00	171.00	2.54–3.38	230	15
Ethanolamine	232.08	171.00	2.84–3.68	120	25
Aspartate	304.00	171.00	3.10–3.94	160	16
Methylamine	202.00	171.00	3.48–4.32	120	25
Glutamate	318.00	171.00	4.45–5.29	210	16
Citrulline	346.20	171.00	4.54–5.38	120	25
Threonine	290.00	171.00	5.12–5.96	120	10
Alanine	260.00	171.00	5.58–6.42	160	13
Ethylamine	216.00	171.00	5.77–6.61	120	25
Proline	286.00	171.00	6.01–6.85	130	15
Ornithine	473.30	171.00	6.64–7.48	120	25
Cysteine	291.00	171.00	7.06–7.90	120	13
Lysine	487.00	171.00	7.25–8.09	230	22
Tyrosine	352.00	171.00	7.30–8.14	210	15
Putrescine	429.15	171.00	7.38–8.22	120	25
Methionine	320.00	171.00	7.52–8.36	120	16
Serotonin ^1^	347.20	171.00	7.58–8.42	120	25
Valine	288.00	171.00	7.63–8.47	100	16
Cadaverine	443.20	171.00	7.65–8.49	120	25
Tyramine	308.20	171.00	7.68–8.52	120	25
Isoleucine	302.00	171.00	8.08–8.92	120	25
Leucine	302.00	171.00	8.08–8.92	120	25
Serotonin ^2^	517.20	171.00	8.14–8.98	130	15
Phenylalanine	336.00	171.00	8.30–9.14	160	20
Tryptophan	375.00	171.00	8.41–9.25	160	20
SIL Tryptophan	388.00	171.00	8.41–9.25	120	25
Tryptamine	331.20	171.00	8.54–9.38	120	25
2-phenethyl-amine	292.20	171.00	8.54–9.38	230	18

## Data Availability

The acquired SRM data are publicly available at the Panorama website: (https://panoramaweb.org/University%20of%20Debrecen/AA_BA_Validation/project-begin.view? accessed on 21 January 2022).

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
