# Peer review of "Fast and Sensitive Quantification of AccQ-Tag Derivatized Amino Acids and Biogenic Amines by UHPLC-UV Analysis from Complex Biological Samples"

_metabolites, 2022, doi:10.3390/metabo12030272_

Round 1

Reviewer 1 Report

This is interesting method development manuscript that is done well. There are couple of minor issues that the authors should address and these points are listed below. There are also several wording issues that need to be cleaned up.

  1. Page 3, line 111: wavelength is misspelled.
  2. Figure 1:  There are several peaks (1.8, 2.2, and 4.7 min.) that are not labelled. Do the authors have any insight as to what the peak are do to?
  3. Page 6, line 192:  The word 'form' should be "from".

Author Response

We would like to thank the reviewer for reading our manuscript and suggesting changes, which can increase the quality of our manuscript.

  1. Page 3, line 111: wavelength is misspelled.
  2. Figure 1:  There are several peaks (1.8, 2.2, and 4.7 min.) that are not labelled. Do the authors have any insight as to what the peak are do to?
  3. Page 6, line 192:  The word 'form' should be "from".

We have corrected the mentioned misspellings and highlighted them with yellow color in the text.

In case of Figure 1, the peak found at 1.8 min is the single derivatized lysine that according to the vendor’s instructions sometimes is produced during the derivatization process. Since the level of the single derivatized form is low compared to the double derivatized form, we have used only the double derivatized form for identification and quantification as it was recommended by the vendor. Peaks at 2.2 min and 4.7 min represents the single derivatized putrescine and cadaverine, respectively. Since the intensity of these single derivatized peaks were constantly lower than the double derivatized peaks and the ratio between the single and double derivatized peaks were similar, we used the double derivatized peaks for identification and quantification.

We have indicated this information in the manuscript between lines 151-155 and highlighted with yellow as following:

“In case of lysine, cadaverine and putrescine we have detected both single and double derivatized peaks but since the intensity of the single derivatized peaks was low and the ratio between the single and double derivatized forms was constant, we have used the double derivatized peaks for quantification.”

Reviewer 2 Report

No comments.

Author Response

We would like to thank the reviewer’s time and effort to reading our manuscript and found it acceptable for publication.

Reviewer 3 Report

This manuscript reported the method development and validation of 33 derivatized amino acid and biogenic amines by UPLC-MS. Under the single injection, all the 33 analytes were well separated, except two. The robustness of the method was further validated by the SRM mode, together with the determination of LOD and LOQ. In addition, the established method was applied to analyze the components from the serum and tears. Well, it is a well-organized metabolites analysis manuscript, with a focus on method development and validation. As far as I am concerned, this manuscript can be assigned as the minor revision.    Here are my questions and concerns.   1. The data from Table 1-3 did not mention the replication info, and I assume the data shown is an average of the several analyses, it is highly recommended to add this information below the table.   2. In the introduction part, it might be necessary to briefly introduce why those specific biogenic amines were selected. There is no need to explain the 20 amino acids, but it is recommended to talk about the biogenic amines used in this manuscript.    3. For those biogenic amines which are under LOQ, do we have a backup method or another way to analyze it? If so, even it is not included in this manuscript, it might be good to give the suggestions and possible plan. This manuscript reported the method development and validation of 33 derivatized amino acid and biogenic amines by UPLC-MS. Under the single injection, all the 33 analytes were well separated, except two. The robustness of the method was further validated by the SRM mode, together with the determination of LOD and LOQ. In addition, the established method was applied to analyze the components from the serum and tears. Well, it is a well-organized metabolites analysis manuscript, with a focus on method development and validation. As far as I am concerned, this manuscript can be assigned as the minor revision.    Here are my questions and concerns.   1. The data from Table 1-3 did not mention the replication info, and I assume the data shown is an average of the several analyses, it is highly recommended to add this information below the table.   2. In the introduction part, it might be necessary to briefly introduce why those specific biogenic amines were selected. There is no need to explain the 20 amino acids, but it is recommended to talk about the biogenic amines used in this manuscript.    3. For those biogenic amines which are under LOQ, do we have a backup method or another way to analyze it? If so, even it is not included in this manuscript, it might be good to give the suggestions and possible solution.

Author Response

We would like to thank the reviewer for reading our manuscript and suggesting changes, which can increase the scientific quality of our manuscript. Please find the answers for your questions below:

  1. The data from Table 1-3 did not mention the replication info, and I assume the data shown is an average of the several analyses, it is highly recommended to add this information below the table.

Thank you very much for pointing out this issue. We have added the necessary information regarding the replications for all three tables in the manuscript highlighted with yellow color.

  1. In the introduction part, it might be necessary to briefly introduce why those specific biogenic amines were selected. There is no need to explain the 20 amino acids, but it is recommended to talk about the biogenic amines used in this manuscript.

 The selection of the biogenic amines for this study was based on their roles in physiological processes and in pathological conditions. We have inserted a new paragraph that describes the role of serotonin, ethylamine, ethanolamine and methylamine between lines 76-88 highlighted with yellow as the following:

“Ethylamine is an organic compound that found to be negatively associated with the risk of the development of type 2 diabetes mellitus in a Japanese population [18]. Methylamine is one of the simplest aliphatic amine in the human body and has been identified in many tissues and body fluids [19]. It has been suggested that methylamine plays a role in central nervous system disturbances observed in several pathological conditions [19] and it has been observed that in pregnancy toxemia, the blood level of methylamine remains higher for a longer time after delivery compared to normal pregnancies [20]. Ethanolamine can be found in every cells in the human body as part of phospholipids, and in a free form in body fluids. As the main component of phosphatidylethanolamine, ethanolamine plays a role in neurodegenerative disorders, cancer and ferroptosis [21]. Serotonin as an important neurotransmitter of the central nervous system has a key role in the development of depression [22] and found to be associated with obesity and diabetes mellitus as well [23].”

We have also inserted a new sentence between lines 106-107 highlighted with yellow as the following:

“The selection of the biogenic amines for this study was based on their roles in physiological processes and in pathological conditions.”

  1. For those biogenic amines which are under LOQ, do we have a backup method or another way to analyze it? If so, even it is not included in this manuscript, it might be good to give the suggestions and possible plan.

Although the concentration of biogenic amines was lower than the limit of quantification in the samples of healthy volunteers, the method has relevance in the analysis of samples from pathological conditions where the level of these molecules can be higher than in healthy conditions. Our obtained result with this small number of samples indicates that we could quantify these molecules in clinical samples and the method can be useful for the analysis of these molecules in studies involving bigger cohorts. We have used this method for the examination biogenic amines in serum and tears originating from patients with diabetes and obesity (manuscript submitted to IJMS). Another possibility to increase the sensitivity of our method for biogenic amines is the increase of the sample volume used for derivatization. This can be easily done in case of serum samples but in case of tears, the volume of the collected basal tear is always limited. However, our method can be tested with 5µl tear sample in order to increase the concentration of the small molecules selected in this study. In addition, the method can be further transferred into a UHPLC-MS system that contains a high-resolution mass spectrometer (e.g. Orbitrap) that provides higher sensitivity for the quantification of biogenic amines.

Reviewer 4 Report

Dear authors,

The manuscript has relevant information and I found some interest in it, but I have some comments.

Comments:

Plagiarism was noted in the following paragraph, please rewrite and check all text. "The identification of biomarkers specific for various pathological conditions is an important field of medical sciences. In some conditions, biomarkers have a central role in normal or pathological functions and their presence or absence causes the malfunction leading to disease [4]. In other cases, the presence, absence or differential expression of the biomarkers is the consequence and not the cause of the disease [5]."

Please remove the words amino acid and biogenic amine from keywords. They are present in the title.

Author Response

We would like to thank the reviewer for reading our manuscript and suggesting changes, which can increase the quality of our manuscript.

Plagiarism was noted in the following paragraph, please rewrite and check all text. "The identification of biomarkers specific for various pathological conditions is an important field of medical sciences. In some conditions, biomarkers have a central role in normal or pathological functions and their presence or absence causes the malfunction leading to disease [4]. In other cases, the presence, absence or differential expression of the biomarkers is the consequence and not the cause of the disease [5]."

Thank you very much for pointing out this issue. We have rewritten the suggested paragraph and we have highlighted it with yellow color as the following:

“The identification of biomarkers specific for different diseases is an important field in biological and medical sciences. In some conditions, the identified biomarkers have a central role in the homeostatic or pathological processes and their presence or absence has a key role in the manifestation of the disease [4]. In other cases, the presence, absence or altered level of the biomarkers is the consequence of the disease with no connection of the cause of the disease [5].”

Please remove the words amino acid and biogenic amine from keywords. They are present in the title.

We have removed the amino acid and biogenic amine from the keywords according to the Reviewer’s suggestion.